# Short-Term Effects of Competitive Video Games on Aggression: An Event-Related Potential Study

**DOI:** 10.3390/brainsci13060904

**Published:** 2023-06-02

**Authors:** Jiayi Sun, Junyi Hao, Yanling Liu

**Affiliations:** 1School of Psychology, Nanjing Normal University, Nanjing 210097, China; 17784413014@163.com (J.S.);; 2Research Center of Mental Health Education, Faculty of Psychology, Southwest University, Chongqing 400715, China

**Keywords:** competitive video game, aggression, event-related potential, P300

## Abstract

Previous research on factors affecting video game player aggression has mainly reflected on the violent content of video games; in recent years, some researchers have focused on competitive factors in video games. However, little research has examined the sole impacts of competitive factors in video games without violent content on aggression, and the neurological processes of these effects are still unknown. The present study was the first to examine the electrophysiological characteristics of short-term competitive video game exposure and aggression. Thirty-five participants played a video game in either competitive or solo mode for 15 min, followed by an ERP experiment based on the oddball paradigm and the hot sauce paradigm to measure aggressive behavior. Results showed that playing competitive game mode was associated with faster judgment of aggressive words, larger P300 amplitudes, and selection of more chili powder than in solo mode. Mediation analysis further revealed that the P300 amplitude evoked by the aggressive words partially mediated the relationship between competitive game exposure and aggressive behavior. These findings support the general aggression model. However, this study has limitations, such as a single form of competitive game examined and single blindness, which need further improvement in future studies.

## 1. Introduction

Video games have become one of the most well-liked kinds of entertainment worldwide, with a projected 3.2 billion players and a market value of USD 196.8 billion in 2022, expected to rise to USD 225.7 billion by 2025 [1]. As video games continue to grow in popularity, researchers have become increasingly interested in their effects on players. Past research has found that violent video games can lead to increased aggression in players [2,3,4,5,6,7,8,9]. However, recent research has begun to question the role of violent content in video games [10,11] and has suggested that competitive factors may have a greater impact on player aggression [12,13,14]. Although the genre of competitive video games is popular [15,16], little research has investigated the effects of competitive factors on aggression in video games without violent content. The purpose of the current study was to explore the short-term effects of competitive video games on aggression and the underlying neural correlates. 

Competition involves a negative correlation between individual goals, where achieving one person’s goal implies that others do not achieve theirs [17]. The general aggression model (GAM) [18,19] proposes that aggression is the result of a dynamic interplay between personal and situational factors. Competition in video games is used as a situational factor to have an impact on players’ aggression. The GAM suggests that input variables (e.g., personal and situational factors) can influence internal states, such as aggressive thoughts, angry emotions, and physiological arousal, which can then lead to aggressive behavior.

In terms of behavioral research, the first study to examine how competitive factors in video games affect aggressive behavior was conducted by Anderson and Morrow in 1995 [20]. However, to date, very limited research has examined competition as a primary factor [13]. Most of the video game materials used in experimental studies that involved examining the effects of competitive factors also contained violent content, such as boxing games [21,22] and shooter games [13,23,24]. These studies found that competitive factors in violent video games increase aggressive cognition [25,26], hostility [13,24,27,28,29], and aggressive behavior [11,14]. One possible explanation for this is that competitive game modes provide incentives (e.g., win or upgrade) for players to use violent strategies, leading to enhanced aggression [26]. However, video games with competitive factors are generally popular [15,16], and many of them do not contain violent content. It is not clear how competitive factors affect aggression in nonviolent video games, where players cannot use violent strategies. To examine the question, a racing game, which is nonviolent but competitive, was selected as the experimental material for this study. 

For why players engage in aggressive behavior after a short exposure to video games, aggression cognition, emotion, and arousal are all explained in the general aggression model [18]. Previous research has primarily focused on cognition as a key factor in video-game-induced aggression [26,30,31]. Therefore, aggressive cognition is a logical starting point for examining the effects of competitive video games on aggressive behavior. Aggressive cognition has often been assessed using reaction times to aggressive words, where shorter reaction times indicate increased aggressive cognition [30,32]. In the video game literature, the competitive reaction time task (CRTT) is commonly employed to measure aggressive behavior [33,34,35]. In this task, participants compete against an opponent, typically the system, to respond faster to a target square on the screen. The loser is then subjected to a noise punishment determined by the opponent, and the intensity and duration of the noise serve as indicators of aggressive behavior. However, concerns about the reliability and validity of the CRTT have been raised [36,37]. Moreover, this study examines the effects of competitive video games. Competitive video games will prime competitive schemas more than noncompetitive video games because the former must entail more competition, which makes the competitive component of the CRTT particularly salient. To avoid possible confusion, the hot sauce paradigm, which more directly and explicitly measures aggressive behavior, was chosen for this study. In this paradigm, participants are tasked with preparing hot sauce for another participant who has expressed a dislike for spicy food. The amount of chili powder added serves as an indicator of aggressive behavior [38]. Previous studies have provided support for the convergent validity of this paradigm [11,36,38].

Studies on the effect of competitive video games on aggression were at the behavioral level, and no studies have been conducted to examine the relationship more directly and effectively at the neural correlates level. Relevant EEG and ERP studies have been conducted in the video game domain, where the P300 component has been commonly involved. The P300 is a positive amplitude that appears approximately 300 ms after the stimulus and is involved in cognitive processing, is used to examine cognitive processes, and is considered an indicator of neural resource allocation, and its amplitude is positively correlated with the mental resources invested [39,40]. The P300 component can be stimulated using the dual-stimulus oddball paradigm. The stimulus consists of two kinds: one is the target (novel) stimulus that appears by chance and the other is the nontarget stimulus that appears frequently. Previous research has found that long-term exposure to violent video games leads to desensitization to violence and that participants with high violent video game experience have a smaller P300 amplitude for viewing violent images compared to participants with low violent game experience, which, in turn, leads to aggressive behavior [41,42]. However, after short-term exposure to violent video games, participants viewing violent images show an increase in the amplitude of P300, showing sensitivity to violence [43]. Studies have also examined the neural correlates of the effects of prosocial video games and found that short-term play of prosocial video games leads to smaller P300 amplitudes for aggressive words, which, in turn, reduces aggressive behavior [44]. Given these findings, it is possible to characterize the neural correlates of competitive video games affecting aggression by observing participants’ P300 amplitudes in response to negative stimuli.

The present study aimed to investigate the effect of competitive factors on aggression in a nonviolent video game using ERP. Two game modes of a nonviolent racing game were selected: a competitive and a solo game play mode. Aggressive cognition was assessed through participants’ reaction time and P300 amplitude to aggressive words using the oddball paradigm, while aggressive behavior was evaluated using the hot sauce paradigm. This study had three research objectives. Firstly, we aimed to examine the effects of competitive mode in nonviolent video games on aggressive cognition and behavior. Secondly, we sought to explore the neural correlates of the effects of competitive video games. Lastly, we aimed to explore the mediating role of aggressive cognition in the effect of competitive game mode on aggressive behavior. According to the GAM and empirical studies, our first hypothesis (H1) was that players would exhibit higher levels of aggressive cognition and behavior in the competitive mode compared to the solo mode. The second hypothesis (H2) stated that participants with short-term exposure to the competitive mode of a racing game would produce larger P300 amplitudes for the aggressive words compared to the solo mode. Given that aggressive cognition is believed to mediate the relationship between video game exposure and aggressive behavior in GAM, our third hypothesis (H3) proposed that P300 amplitude for aggressive words would mediate the effect of the competitive game mode on aggressive behavior. 

## 2. Materials and Methods

### 2.1. Participants

The sample size of 34 participants was calculated using G*Power 3.1 [45] for the repeated-measures factorial ANOVA, achieving a statistical power of 0.80 with a significance level of 0.05 and a medium effect size *f* = 0.25 [46,47]. A total of 40 college students (21 women and 19 men, aged 19–21 years, *M* = 20.01, *SD* = 0.91) with normal vision and no neurological disease or brain injury were recruited from a university in Chongqing, Southwest China. Participants were randomly assigned to either the competitive or solo game play mode, with 20 participants in each group. After excluding six participants due to large data artifacts, the final number of valid participants was 19 in the competitive mode (8 women and 10 men) and 16 in the solo mode (10 women and 7 men). Before EEG data collection, participants were instructed to minimize the number of blinks and avoid irrelevant body movements, such as leg shaking. The study was approved by the Human Research Ethics Committee of Southwest University in China (H19001).

### 2.2. Material and Measures

#### 2.2.1. Video Games

The present study utilized the QQ Speed racing game, which offers two distinct game modes: competitive and solo game play. To control for potential confounders resulting from differences in games [48], we selected different modes of the same game instead of using two different games. In the solo mode, participants drove the game’s car for a level-breaking task for 15 min while performing noncompetitive activities as instructed. Players earned points for completing each level, and their scores were displayed at the end of each level and the end of the game. In the competitive mode, participants competed against five other computer players in a race to cross the finish line first. The game was played over three rounds, each lasting 15 min. Participants’ scores and rankings in the race were displayed on the screen at the end of each round, and the researchers recorded the top ranking achieved in the previous three rounds. Previous research has shown that the presence of opponent scores enhances competitive behavior [49]. Participants in the solo mode received a fixed payout of CNY 50, while those in the competitive mode were compensated based on their rankings, up to CNY 5 for the first-place finisher. All participants completed the game on identically configured computers in the same laboratory.

An adaptation of the Game Evaluation Questionnaire [50] was administered after the game to assess participants’ perceptions of the game. The questionnaire comprises seven dimensions: aggressive and prosocial content, competition, actions, enjoyment, excitement, and difficulty. Participants rated these seven dimensions on a five-point Likert scale ranging from 1 (not at all true of me) to 5 (very true of me) based on their experience of playing the game. 

#### 2.2.2. Aggressive Cognition 

An oddball paradigm was used in which participants judged whether the presented word stimuli were aggressive words or not. In this paradigm, aggressive words were the target (novel) stimuli, and an aggressive word trial was presented after every 3 to 5 neutral word trials, with the number of aggressive word trials accounting for 20% of the total number of trials. First, a “+” symbol was presented for 200 ms, followed by a black screen for 500 ms. Subsequently, a stimulus presentation and response screen appeared for 1000 ms. Participants judged aggressive words by pressing the “n” key and did not respond to neutral words. The stimulus remained on a time-randomized black screen (750–1350 ms) after the judgment, while the participant’s response was recorded. The task had two blocks of 150 trials each and was presented in the middle of the computer screen using E-prime software (Psychology Software Tools Inc., Sharpsburg, MD, USA). Participants completed 20 practice trials before the formal task, with a break between each block. 

The words used for this task were determined by a preliminary experiment. A total of 30 aggressive words and 60 neutral words were selected from the Modern Chinese Dictionary. Thirty participants who did not participate in the formal experiment rated these words on a scale of 1−5 on three dimensions, including universality, understandability, and aggression. There was no difference in universality, *t*_(88)_ = 0.16, *p* = 0.87, 95% CI [−0.19, 0.22], *d* = 0.04, 95% CI [−0.40, 0.47], between the aggressive (*M* = 3.54, *SD* = 0.21) and neutral words (*M* = 3.52, *SD* = 0.54). There was also no difference in understandability, *t*_(88)_ = 0.07, *p* = 0.95, 95% CI [−0.18, 0.19], *d* = 0.02, 95% CI [−0.42, 0.45], between the aggressive (*M* = 4.21, *SD* = 0.24) and neutral words (*M* = 4.20, *SD* = 0.48). A significant difference was found only in aggression, *t*_(88)_ = 40.74, *p* < 0.001, 95% CI [2.78, 3.06], *d* = 9.11, 95% CI [7.69, 10.50], with aggressive words (*M* = 4.21, *SD* = 0.44) being significantly more aggressive than neutral words (*M* = 1.29, *SD* = 0.25).

#### 2.2.3. Aggressive Behavior

Aggressive behavior was measured using the hot sauce paradigm [11,51]. Participants first indicated their preferred level of spiciness and then were instructed to add chili powder to a hot sauce concoction for another participant who had expressed a dislike for hot meals. Real chili powder was used to enhance the paradigm’s authenticity, and varying amounts of chili powder were placed in clear cups labeled from 1 (a little hot) to 5 (very hot). The amount of chili powder selected for another participant was used as an indicator of aggressive behavior.

#### 2.2.4. Trait Aggression

Trait aggression was measured using the Aggression Questionnaire [52], a validated questionnaire for measuring aggressive traits [50]. The questionnaire contains 29 items on a five-point Likert-type scale ranging from 1 (does not reflect how I feel at all) to 5 (accurately reflects how I feel). The internal consistency of this questionnaire in this study was 0.85. There was no difference in trait aggression, *t*_(33)_ = −0.43, *p* = 0.68, 95% CI [−0.47, 0.31], *d* = −0.14, 95% CI [−0.81, 0.52], between the competitive (*M* = 2.56, *SD* = 0.67) and solo mode groups (*M* = 2.47, *SD* = 0.41), indicating a match between two groups on the trait aggression.

#### 2.2.5. Video Game Exposure

Participants’ exposure to both violent and competitive video games was measured. Violent video game exposure was measured using the Video Game Experience Questionnaire [50]. Participants were asked to list the three video games they had been playing often during the previous six months, along with the frequency of use (1 = seldom, 7 = frequently) and the rating level of violence in each game (1 = none, 7 = very much). Violent video game experience was calculated using the formula: Ʃ [degree of violence of game content × (frequency of playing games on weekdays × 5 + frequency of playing games on weekends × 2)/7]/3 [53]. Previous studies have demonstrated the validity and reliability of this self-report measure of violent game exposure [54]. This method of measuring has been extensively employed in earlier studies, including those conducted in China [5,55,56]. In accordance with the study, we added a measure of exposure to competitive video games in reference to the measure of violent game exposure. Participants filled in their frequency of use and rating of the level of competition for the three games they played regularly in the last six months on the same 7-point scale. Competitive video game experience was calculated using the formula: Ʃ [degree of competition of game content × (frequency of playing games on weekdays × 5 + frequency of playing games on weekends × 2)/7]/3. There was no difference in violent video game experience, *t*_(33)_ = −0.60, *p* = 0.55, 95%CI [−6.16, 3.34], *d* = −0.21, 95%CI [−0.87, 0.46], between the competitive (*M* = 10.94, *SD* = 7.19) and solo mode groups (*M* = 9.53, *SD* = 6.48). There was also no difference in competitive video game experience, *t*_(33)_ = −1.20, *p* = 0.24, 95%CI [−8.38, 2.16], *d* = −0.41, 95%CI [−1.08, 0.27], between the competitive (*M* = 17.14, *SD* = 8.46) and solo mode groups (*M* = 14.03, *SD* = 6.50). The results showed that both groups of participants were matched on both violent and competitive video game experiences.

### 2.3. Procedure

Participants were screened to ensure that they had no or minimal experience with the experimental game QQ Speed. Upon arrival at the laboratory, participants first signed an informed consent form. They were informed that they would be evaluating a video game and participating in an ERP experiment examining attention. Participants then filled out the Aggression Questionnaire and the Video Game Experience Questionnaire. After this, participants had an EEG electrode cap placed on their heads and electrode impedances were reduced. Next, participants were randomly assigned to play either the competitive or solo mode of QQ Speed for 15 min after a 5-min practice session. Participants completed the Game Evaluation Questionnaire following the game. After a two-minute rest, EEG recording started and participants completed an oddball paradigm and the hot sauce paradigm. After completing all tasks, participants were asked to report their perceived purpose of the experiment they had just participated in. A 5-min prosocial video was shown to participants at the end of the experiment to avoid initiating aggression.

### 2.4. Electrophysiological Recording and Data Processing

EEG data were recorded using 64-channel 10–20 system EEG caps with active Ag/AgCl electrodes (Brain Products, Gilching, Germany). A vertical electrooculography (VEOG) was recorded below the midpoint of the right eye, and a horizontal electrooculography (HEOG) was recorded from the outer canthi of the right eye. The electrode impedances were maintained below 5 kΩ prior to the start of recording. EEG data were analyzed using MATLAB R2016a (MathWorks, Inc., Natick, MA, USA) and the EEGLAB toolbox [57]. EEG signals were amplified in a 0.01–30 Hz bandpass and resampled at 500 Hz. The EEG was re-referenced to the average of the bilateral mastoids. Independent component analysis (ICA) was used to remove eye movements, eye blinks, and other artifacts. Time windows of 200 ms before and 800 ms after the onset of word presentation were segmented from the EEG data. The whole epoch was baseline-corrected by the 200-ms time interval before the onset of the words. Epochs containing incorrect responses or amplitude values exceeding ±75 μV at any electrode were excluded from the final average. Based on previous ERP studies studying video games [42,44] and voltage scalp topographies (Figure 1) in the current results, the positive component from 300 to 800 ms over the parietal area (P3, Pz, P4) was analyzed as P300. 

Data were analyzed using SPSS version 22 (SPSS Inc., Chicago, IL, USA). Manipulation checks and behavioral data were conducted using independent samples *t*-tests to examine the differences between the two game play modes. The average P300 amplitude was analyzed using a 2 (game play mode: competitive vs. solo) × 2 (word type: aggressive vs. neutral) repeated-measures factorial ANOVA. The mediation analysis was performed using the PROCESS macro for SPSS. Model 4 [58] was used to examine the mediation effect of aggressive cognition (i.e., P300 amplitude in response to aggressive words) on the influence of game play mode on aggressive behavior. Analyses used 5,000 bootstrapping samples and 95% bias-corrected confidence intervals to examine the significance of estimated effects [59].

## 3. Results

### 3.1. Manipulation Check

An independent samples *t*-test was performed to compare the two modes regarding game evaluation (see Table 1). The results indicated a significant difference between the two modes only in the competition dimension, *t*_(33)_ = −3.71, *p* < 0.001, 95% CI [−1.73, −0.50], *d* = −1.26, 95% CI [−1.98, −0.52], with a significantly higher degree of competition in the competitive mode than in the solo mode. There were no significant differences between the two modes in the other dimensions (all *p*> 0.10). These results suggest that participants could differentiate between the competitive and solo modes and that the two modes were comparable in the other dimensions.

### 3.2. Behavioural Data

#### 3.2.1. Oddball Paradigm

The reaction time (ms) for aggressive words differed significantly between the competitive and solo modes, *t*_(33)_ = 2.87, *p* = 0.007, 95% CI [16.83, 98.96], *d* = 0.98, 95% CI [0.26, 1.67]. Specifically, participants in the competitive mode (*M* = 516.72, *SE* = 13.35) responded significantly faster to aggressive words than those in the solo mode (*M* = 574.62, *SE* = 15.26). However, there was no significant difference in the correctness of responses to aggressive words between the two modes, *t*_(33)_ = −1.14, *p* = 0.26, 95% CI [−0.016, 0.004], *d* = −0.39, 95% CI [−1.05, 0.29]. The mean correctness was 0.993 (*SE* = 0.003) for the competitive mode group and 0.988 (*SE* = 0.004) for the solo mode group.

#### 3.2.2. Hot Sauce

There was a significant difference in aggressive behavior between the competitive and solo modes, *t*_(33)_ = −3.28, *p* = 0.002, 95% CI [−1.74, −0.41], *d* = −1.11, 95% CI [−1.82, −0.39]. The hot sauce score was higher for the competitive mode (*M* = 2.95, *SE* = 0.23) than for the solo mode (*M* = 1.88, *SE* = 0.22). A sensitivity analysis was conducted by G*Power 3.1 [45] for independent samples *t*-test with α= 0.05. We had 80% power to detect an effect size of *d* = 0.97, smaller than our observed effect size of the oddball paradigm and hot sauce results.

### 3.3. ERP Results

Figure 1 shows the grand average waveforms elicited by the words for every condition in the two game play modes. The main effect of word type was statistically significant, *F*(1,33) = 150.79, *p* < 0.001, η^2^*_p_* = 0.82. Specifically, P300 amplitudes were larger for aggressive words (*M* = 5.65, *SE* = 0.50) than for neutral words (*M* =1.30, *SE* = 0.29). The main effect of game play mode was also statistically significant, *F*(1,33) = 5.71, *p* = 0.02, η^2^*_p_* = 0.15. Participants in the competitive mode (*M* = 4.36, *SE* = 0.50) had larger P300 amplitudes than those in the solo mode (*M* =2.59, *SE* = 0.55). Furthermore, the interaction between game play mode and word type was also significant, *F*(1,33) = 5.27, *p =* 0.03, η^2^*_p_* = 0.14. A simple effects test was conducted, which revealed that, for aggressive words, game play mode differed significantly, *F*(1,33) = 6.63, *p* = 0.015, η^2^*_p_* = 0.17. Specifically, P300 amplitudes for aggressive words were significantly larger in the competitive mode (*M* = 6.94, *SE* = 0.68) than in the solo mode (*M* = 4.35, *SE* = 0.74). However, for neutral words, game play mode differences were not significant, *F*(1,33) = 2.69, *p =* 0.11, η^2^*_p_* = 0.08.

### 3.4. Mediation Model Analysis

A mediation model was conducted with video game play mode (0 = solo; 1 = competition) as an independent variable (X), aggressive cognition (i.e., P300 amplitude in response to aggressive words) as mediator (M), and aggressive behavior (hot sauce scores) as a dependent variable (Y) (Figure 2). The results revealed that the direct effect of game play mode on hot sauce scores was significant (β = 0.66, *SE* = 0.33, 95% CI [0.49, 1.40]). The game play mode significantly predicted larger P300 amplitude evoked by aggressive words (β = 0.81, *SE* = 1.01, 95% CI [0.54, 4.64]), and larger P300 amplitude significantly predicted increased hot sauce scores (β = 0.39, *SE* = 0.05, 95% CI [0.03, 0.24]). Furthermore, P300 amplitude evoked by aggressive words partially mediated the relationship between game play mode and hot sauce scores (β = 0.32, Boot *SE* = 0.20, 95% Boot CI [0.01, 0.79]). The effect ratio was 0.32, indicating that aggressive cognition mediated approximately 32% of the relationship between competitive game mode and aggressive behavior.

## 4. Discussion

The present study aimed to investigate the effects of competitive video games on aggression at the behavioral and neural levels. The results showed that players in the competitive game mode reacted to aggressive words significantly more quickly and chose more chili powder than those in the solo mode, supporting H1. Consistent with previous findings, competitive factors in video games can increase aggressive cognition [25,26] and aggressive behavior [11,12,14,20]. The results support the short-term effect of the general aggression model, which suggests that situational factors (competitive video games) influence individual aggression [18]. Notably, instead of using a violent video game, which has been used as experimental game material in most previous related studies [13,21,22,23,24], this study chose a racing game that did not contain violent content but was competitive. The study found that the competitive mode in the nonviolent video game still significantly affected aggression, which extends the existing research. Activation of scripts associated with aggression in the face of competition may explain this effect [20], as competitive interactions in video games, even without violent content, may trigger memories of hostility and aggression. The influence of competition on aggression is also strongly supported by Deutsch’s competition effect theory in social psychology [17,60].

The current study was the first to examine the relationship between competitive video games and aggression at the level of neural correlates. Our findings revealed that P300 amplitudes were larger for aggressive words than neutral words, consistent with previous research suggesting that target stimuli elicit larger P300 amplitudes than nontarget stimuli [61,62]. The oddball paradigm of this study successfully induced the P300 component. Furthermore, we discovered a significant interaction between game play mode and word type, with P300 amplitudes for aggressive words being significantly larger in the competitive game play mode than the solo mode, supporting H2. Similar to the findings of neural correlates of the effects of short-term exposure to violent video games on aggression [43,63], the results of this study suggest that, after short-term exposure to competitive video games, players allocate more attentional resources to aggressive cues and exhibit sensitivity to aggression.

The mediation analysis revealed that the P300 amplitude in response to aggressive words partially mediated the relationship between game play mode and hot sauce scores, supporting H3. According to the GAM, the situational input variable (exposure to competitive video games) contributes to aggressive behavior through aggressive cognition, negative affect, and arousal [64]. Of the three pathways, previous research on violent video games has focused on cognition, noting that aggressive cognition plays a crucial role in explaining aggressive behavior [26]. The present study supported that aggressive cognition can partially explain the role of competitive video game exposure on aggressive behavior. However, many studies have focused on the facilitative effects of short-term exposure to violent video games on aggressive affect and arousal [2,3,8]. It remains an intriguing avenue for future research to explore whether competitive video games also impact aggressive affect and arousal and whether these two pathways play mediating roles.

The practical implications of this study are significant, as they shed light on the importance of considering the effects of competitive game modes on players’ aggression. Regarding social policy, the findings urge researchers to exercise caution when making causal inferences about the relationship between video game violence and aggression, thereby enhancing public policy [65]. Within the video game industry, game developers should be mindful of the potential effects of competitive modes and carefully consider their inclusion in game design. Furthermore, the study’s findings have important implications for educational interventions. Educators, particularly those working with children and young adults, should be aware of the potential influence of competitive video game exposure on aggression and take steps to educate students on healthy and appropriate gaming behaviors. Teaching students to recognize and effectively regulate their cognition and behaviors is crucial.

In addition, this study had some limitations. First, the competitive mode was created through game ranking, and other forms of competition may have different effects on aggression [66], so the generalization of the results may be limited. It is essential for future research to test similar hypotheses across a broader range of competitive video game forms. Second, although efforts were made to control dimensions such as actions, enjoyment, and difficulty in both game modes, other aspects, such as rewards, were not controlled. Previous studies have indicated that rewards in violent video games can increase aggression [31,67]. External rewards, such as monetary incentives, can also influence participants’ willingness to engage in gaming [68]. This study did not implement explicit control over in-game and monetary rewards, making perceived rewards a potential confounding factor. Future research should consider controlling for rewards and other relevant variables. Third, because of the limited funding, this study included only 35 participants and the statistical power may be small, so the results should be viewed cautiously. Moreover, the researchers involved in this study were not blind to the experimental conditions, potentially introducing subjective bias into the experimental outcomes. Increasing the sample size to enhance statistical power and conducting more rigorous double-blind experiments are possible improvements for future studies. Finally, this study only examined short-term effects, and longitudinal investigations are necessary to determine whether competitive video games increase aggression in the long term. It is also important to consider differences across age groups. For example, a study indicated that children between the ages of 8 and 11 who played competitive video games displayed fewer conduct-disorder-related behaviors [69].

## 5. Conclusions

The present study utilized behavioral and neural indicators to examine the effects of competitive factors in video games without violent content on player aggression. This study found that short-term exposure to the competitive play mode in nonviolent video games can directly promote players’ aggressive cognitions (shorter reaction times and larger P300 amplitudes for aggressive words) and aggressive behaviors and also indirectly increase aggressive behaviors by activating aggressive cognitions (P300 amplitudes for aggressive words). The results support the general aggression model and contribute to understanding the relationship between video games and player aggression. Amidst the ongoing controversy surrounding the effects of violent video games on aggression, this study provides empirical evidence supporting the notion that competitive factors in video games can elicit aggression, even in the absence of violent content.

## Figures and Tables

**Figure 1 brainsci-13-00904-f001:**
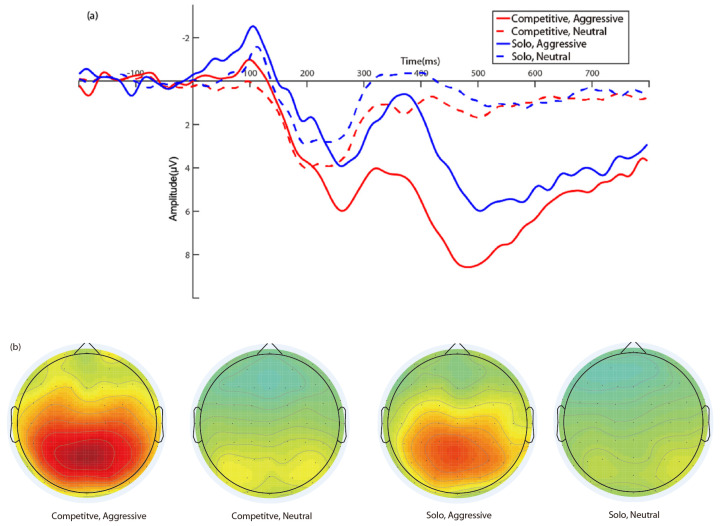
Grand average waves and voltage scalp topographies of P300. (**a**) The grand average waveforms for aggressive and neutral words in competitive and solo game play modes. (**b**) The voltage scalp topographies for aggressive and neutral words in competitive and solo game play modes.

**Figure 2 brainsci-13-00904-f002:**
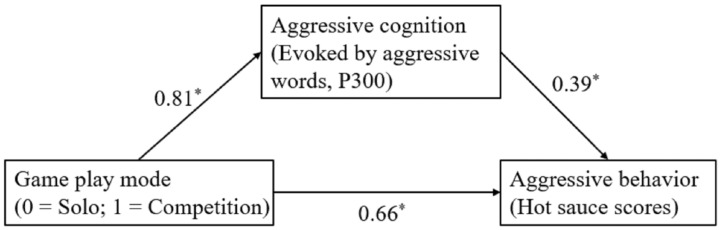
Mediation model of game play mode on aggressive behaviors through aggressive cognition (P300). Standardized path coefficients are shown. Solid lines represent significant paths. * *p* < 0.05.

**Table 1 brainsci-13-00904-t001:** T-test results for the dimensions of game play modes (*n* = 35).

Dimension	Solo Mode *M (SD)*	Competitive Mode *M (SD)*	*t* _(33)_	*p*	95% CI for Mean Difference	*d*	95% CI for *d*
Aggressive content	1.75 (0.93)	1.68 (0.95)	0.21	0.84	[−0.58, 0.71]	0.07	[−0.60, 0.74]
Prosocial content	2.06 (1.12)	2.26 (0.93)	−0.58	0.57	[−0.91, 0.51]	−0.20	[−0.86, 0.47]
Competition	2.94 (1.06)	4.05 (0.71)	−3.71	< 0.001	[−1.73, −0.50]	−1.26	[−1.98, −0.52]
Actions	3.94 (0.77)	3.90 (0.99)	0.14	0.89	[−0.58, 0.66]	0.05	[−0.62, 0.71]
Enjoyment	2.94 (1.18)	3.37 (0.83)	−1.26	0.22	[−1.13, 0.26]	−0.43	[−1.10, 0.25]
Excitement	3.31 (0.95)	3.53 (0.91)	−0.68	0.50	[−0.85, 0.42]	−0.23	[−0.90, 0.44]
Difficulty	2.94 (1.06)	3.42 (1.12)	−1.30	0.20	[−1.24, 0.27]	−0.44	[−1.11, 0.24]

## Data Availability

Publicly available datasets were analyzed in this study. These data can be found here: [https://osf.io/48vhs/?view_only=b17b75783cb744b1ac89a0f69719475f] (accessed on 16 May 2023).

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
