# Peer review of "Short-Term Effects of Competitive Video Games on Aggression: An Event-Related Potential Study"

_brainsci, 2023, doi:10.3390/brainsci13060904_

Round 1

Reviewer 1 Report

In the current study, the authors test whether competition in a non-violent video game increases aggressive behavior via activation of aggressive cognition, measured by behavior and neural responses (P300) to aggressive words in an oddball paradigm. The results largely support their hypotheses, providing additional support to the General Aggression Model and adding to our understanding of video game effects.

The merits of the study are 1) a simple design that tests a direct hypothesis from the GAM, and 2) a focus on competition rather than violence in video games. There are some limitations that would need to be addressed before this study is suitable for publication. Some of these may have to do with a language barrier. I am listing the necessary revisions in order of appearance in the manuscript.

Abstract, lines 11-12: The sentence that begins “However,” needs grammatical revision.

Introduction

Pg 2, Lines 44-54: You use boxing as an example of a violent video game; just a line or two later, you mention "sports" as a non-violent genre of games. Boxing is a sport, so these characterizations are not compatible. In fact, many sports, especially those turned into video games, include some elements of violence, so calling the "sports" genre nonviolent is inaccurate.

Similarly, you list the “strategy” genre as nonviolent, but many of the most popular strategy video games (e.g., Supreme Commander, Total War series, etc.) contain violence.

My view on this issue is that "violent content" and "genre" are separate characteristics of the game. Sometimes there may be a 1:1 relationship between the two (i.e., all first-person shooter genre video games contain violence), but more frequently there is not a perfect correlation (i.e., some strategy video games use violence, some do not). I suggest revising this paragraph so that the two are not conflated. Genre and violent content would need to be independently controlled for or manipulated, depending on the study design, to understand their effects on aggression. It is not appropriate to use the genre as a proxy for identifying how competitive or how violent a video game is.  

Page 2, line 56: Given the racing video game chosen for the study, it is probably fair to call this nonviolent. However, even in this genre there are games that explicitly encourage violence against other players or the computer (e.g., MarioKart, Twisted Metal series). So, I still think you are conflating violence and genre and need to clarify what you really mean by both. It is fair to say you chose a particular racing game that was nonviolent but competitive; it may not be appropriate to call racing video games a nonviolent genre.

Page 3, line 96: Grammar needs edited. I think you are missing a verb in that sentence.

Method

Page 3, line 110: What were the activities that players completed in solo mode? Were the modes similar in terms of rewarding/reinforcing the players for achieving a goal? If not, this could be a potentially important confound to discuss in limitations. I note that you do have the players rate the two modes on certain variables/content, which I think is good and shows they are somewhat similar, but I notice that “rewarding” was not one of the characteristics and this is the one that I would be most concerned about. Can they “win” the solo mode? Do they know how they place in it like how they can tell how they are performing in the competition mode?

Later on, you note the difference in pay for participants completing the two conditions, so even if the game modes themselves were similar in “reward”, the compensation introduced a difference in reward value. Is it possible that how rewarding the game modes felt played an important role in how they felt after playing? Should this at least be addressed as a possible confound in the discussion?

Page 3, line 127: Typically, in a lexical decision task, the participants classify all stimuli as words or nonwords. That does not seem to be the case here, so can the authors explain how this is a lexical decision task?

I do see the oddball element - the aggression words are less frequent. But it sounds like all of the stimuli were real words and participants only had to respond by a button press when an aggressive word appeared. I think you can accurately call it an oddball paradigm, but not a lexical decision task.

Page 4, line 170: How was degree of violence and degree of competition determined? Was it based exclusively on the genre of the games? As I mentioned before, genre would not be a successful marker of violence, by default, so hopefully other markers were used.

Results

Page 5, line 230: See my comment above about how this is a lexical decision task. If you change the name of the task, you’ll need to change the title of this subsection.

Page 5, line 233: There are no units given for the response times. I would assume milliseconds given the averages, but this should be identified.

Page 5; line 239: The section on Hot Sauce is simply the task results copied again. The hot sauce results are not reported, except for their mention in the mediation model.

Discussion

I appreciate that the discussion was concise, clear, and made mention of some limitations. I don’t have many changes to suggest here other than the fact that if the rewarding nature of the two game modes was confounded with the competition of the two modes, then this really should be listed as a limitation of the study.

I noted several grammatical errors throughout the paper. I also wonder if the conflating of genre and violent content in the introduction is also related to how well the English words are understood (i.e., are they familiar with boxing as a sport?). That being said, most of the paper was very easy to understand and I think the language issues can easily be addressed in a revision. 

Reviewer 2 Report

Dear reviewer and authors. I consider the article to be very well developed and well worked. There are some possible improvements, for example, to add some more meta-analysis and some more updated references. Although the objective and hypotheses are not clear to me, I have understood them after reading the article and I see it as coherent. Please review this aspect and rewrite it. On the other hand, in the discussion section, add a section on practical applications, especially with regard to educational aspects.

Reviewer 3 Report

1. Need to proofread - found two errors in abstract [misuse of "solely", should be "sole" and repeated word "the the"

2. Abstract fine, but should add limitations and future directions

3. Introduction clear and logical - need to add a paragraph re evaluation of measures of aggression [eg hot sauce paradigm]

4. Method - all good

5. Results - 3.2.2 section is a repeat of 3.2.1 and thus the hot sauce results are not present

6. Discusion - expected more - especially relate findings to past literature, discuss validity of measures, mention more limitations [researchers not blind to experimental conditions. etc], future research should be discussion and not conclusion, need to discuss implications for the industry or society

7. Conclusion is just a summary - room for a stronger conclusion and recommendations

8. References appear fine

Overall, English quality fine, only minor revisions

Reviewer 4 Report

The manuscript was well written. There were a few minor grammatical issues; however, I think most could be cleaned up without much issue.

Round 2
